# Risk and Prognostic Factors for Glaucoma Associated with Peters Anomaly

**DOI:** 10.3390/jcm12196238

**Published:** 2023-09-27

**Authors:** Chika Yokota, Kazuyuki Hirooka, Naoki Okada, Yoshiaki Kiuchi

**Affiliations:** 1Department of Ophthalmology and Visual Science, Graduate School of Biomedical Sciences, Hiroshima University, Hiroshima 734-8551, Japan; kazuyk@hiroshima-u.ac.jp (K.H.); naokimed@hiroshima-u.ac.jp (N.O.); ykiuchi@hiroshima-u.ac.jp (Y.K.); 2Department of Ophthalmology, Federation of National Public Service and Affiliated Personnel Mutual Aid Associations, Yoshijima Hospital, Hiroshima 730-0822, Japan

**Keywords:** Peters anomaly, glaucoma, congenital corneal opacity, anterior segment dysgenesis, corneal staphyloma

## Abstract

Glaucoma secondary to Peters anomaly is an important factor affecting visual prognosis, but there are few reports on the condition. This study aimed to investigate the characteristics of glaucoma associated with Peters anomaly and glaucoma surgery outcomes. This retrospective study included 31 eyes of 20 patients with Peters anomaly. Peters anomaly was classified into three stages: Stage 1, with a posterior corneal defect only; Stage 2, a corneal defect with iridocorneal adhesion; and Stage 3, a corneal defect with lens abnormalities. The associations between glaucoma and anterior segment dysgenesis severity, visual prognosis, and glaucoma surgery outcomes were analyzed. Sixteen eyes of ten patients developed glaucoma. Stage 1 Peters anomaly had no glaucoma, 52% of Stage 2 had glaucoma, and 75% of Stage 3 had glaucoma. Of the 16 eyes with glaucoma, 11 underwent surgery. Eight of these eleven eyes achieved intraocular pressure (IOP) control. Five of the nine eyes that underwent trabeculotomy (TLO) succeeded, and none had corneal staphyloma. Three of the four eyes for which TLO was ineffective had corneal staphyloma (*p* = 0.0331). Patients with Peters anomaly are more likely to develop glaucoma as anterior segment dysgenesis progresses, and the effect of TLO is limited if corneal staphyloma is present.

## 1. Introduction

Peters anomaly is one of the causes of congenital corneal opacity and was originally described in 1906 by Dr. Alfred Peters, a German ophthalmologist [1]. Peters anomaly is characterized by corneal opacity due to defects in the corneal endothelium and Descemet’s membrane, accompanied by iridocorneal or lenticulocorneal adhesions [2,3]. The peripheral cornea is usually relatively clear, and varying degrees of clouding may accompany the central opacification. Congenital corneal opacities are due to genetic, glaucomatous, infectious, traumatic, developmental, metabolic, idiopathic, or toxic. One of the main causes of congenital corneal opacity is anterior segment dysgenesis [2]. Peters anomaly is estimated to be the most prevalent disease-causing anterior segment dysgenesis with corneal opacity, occurring at an incidence rate of 1.1–1.5 cases per 100,000 people [3]. Homeotic genes controlling the differentiation of primordial cells are responsible for Peters anomaly. Damage to several homeotic genes (*B3GLCT*, *PAX6*, *PITX3*, *FOXE3*, and *CYP1B1*) has been reported to cause Peters anomaly. Notably, most cases of Peters anomaly are solitary; however, autosomal recessive and dominant inheritance patterns have also been reported [4,5]. Homeotic genes are involved in the development of the eye and other body structures; therefore, Peters anomaly is sometimes accompanied by systemic disease. Systemic involvement may present as head anomalies, facial dysmorphism, and ear abnormalities [2]. Approximately half the patients with Peters anomaly have concurrent glaucoma [6,7]. The mechanism of the glaucoma secondary to Peters anomaly is only partially understood. The pathogenesis of glaucoma secondary to Peters anomaly is thought to result from trabeculodysgenesis or angle closure mechanisms associated with iridocorneal and lenticulocorneal adhesions [7]. In the series of Townsend et al., five of the fourteen patients with Peters anomaly had glaucoma, and all showed peripheral anterior synechiae except one with aniridia [8]. On the other hand, Kupfer et al. reported that the mechanism of the noted glaucoma is similar to that of primary open-angle childhood glaucoma due to premature aging changes in the trabecular meshwork [9]. The intraocular pressure (IOP) noted in glaucoma associated with Peters anomaly, among other types of childhood glaucoma, is considered one of the most difficult to control and often requires multiple surgical interventions [10]. Previous studies reporting outcomes of Peters anomaly have focused on penetrating keratoplasty. There have been very few reports on the features and treatment of glaucoma secondary to Peters anomaly due to the difficulty of treatment and the rarity of the disease. This study aimed to describe the characteristics of glaucoma associated with Peters anomaly as well as the treatment outcomes and prognostic factors.

## 2. Materials and Methods

### 2.1. Study Design and Participants

In this study, we reviewed the records of all patients diagnosed with Peters anomaly at the Department of Ophthalmology, Hiroshima University Hospital, between August 2009 and September 2021 and followed up for at least 6 months. The associations between glaucoma onset and the stage of anterior segment dysgenesis, visual outcomes, and postoperative results of glaucoma surgery were retrospectively examined. We collected data from electronic medical records, including age, sex, ocular findings, systemic findings, surgical course, and intraoperative and postoperative complications. This study was conducted according to the principles of the Declaration of Helsinki and was approved by the Institutional Research Ethics Committee (No. E2113). All treatment procedures also followed the principles of the Declaration of Helsinki.

### 2.2. Peters Anomaly Diagnosis and Classification

Peters anomaly was diagnosed clinically and classified as Stages 1, 2, or 3 according to the severity of anterior segment dysgenesis. Stage 1 was defined as the presence of a posterior corneal defect only; Stage 2 as a posterior corneal defect with corneal iris adhesion; and Stage 3 as a posterior corneal defect with lens abnormalities, including lenticulocorneal adhesion (Figure 1). Recently, patients with corneal iris adhesion and central corneal opacity are often classified as having type 1 Peters anomaly. In comparison, those with central corneal opacity associated with adhesion between the cornea and lens have been classified as having type 2 Peters anomaly [5,11]. However, this study classified the degree of anterior segment abnormality into Stages 1–3 to assess glaucoma management more accurately. In addition, cases of Peters anomaly with accompanying systemic disease were classified as Peters plus syndrome. Corneal staphyloma refers to the thinning of the cornea with a focal bulge.

### 2.3. Glaucoma Diagnosis, Postoperative Evaluation

The diagnosis of glaucoma was based on IOP, changes in optic disc corneal diameter, and intraocular axis length. Cases in which the IOP was ≥21 mmHg on more than two occasions and those in which the corneal diameter or ocular axis length was increased were diagnosed with glaucoma [12]. Most Peters anomaly cases were diagnosed in infancy. We could not observe the optic disc in some participants because they also had corneal opacities. Visual acuity (VA) and visual field measurements were impossible at the initial presentation. Furthermore, most participants underwent optotype (Landolt ring) testing at the final follow-up to evaluate VA. IOP was measured mainly with the Icare (Revenue, Vantaa, Finland), Tono-pen (Reichert, Depew, NY, USA), and Perkins applanation tonometers (Haag-Streit UK, London, UK) when a close examination was performed under general anesthesia. Palpation was also performed because it was difficult to accurately measure IOP in cases with strong corneal deformation. Ocular axis length measurement, gonioscopic examination, ultrasound biomicroscopy, and corneal diameter measurement were performed under general anesthesia. This study defined successful glaucoma surgery as a postoperative IOP < 21 mmHg or Tn (normal tension by palpation) without ocular complications that could affect visual function.

### 2.4. Glaucoma Surgical Procedures

Glaucoma surgery was performed when the IOP was >21 mmHg or in cases of corneal diameter or ocular axis length increases. External trabeculotomy (TLO) was selected as the first-line surgery for glaucoma secondary to Peters anomaly, and trabeculectomy (TLE) with mitomycin and glaucoma drainage device (GDD) implantation was performed when TLO was ineffective. The surgical procedures were performed using previously described standard techniques [13,14,15,16]. For TLO, a 4 mm × 4 mm scleral flap with a four-fifths sclera thickness was created to identify the Schlemm canal. Then, a hairpin trabeculotome was used to make a 120° incision in the inner wall of the Schlemm canal. The selected GDDs were the Baerveldt glaucoma implant (BG 101-205, Johnson & Johnson Vision, Santa Ana, CA, USA) in one of the four pediatric patients, Ahmed FP8 (New World Medical, Rancho Cucamonga, CA, USA) in the other three pediatric cases, and Ahmed FP7 (New World Medical, Rancho Cucamonga, CA, USA) in one adult case. A GDD tube was inserted approximately 2 mm from the limbus through a short tunnel and placed in the anterior chamber parallel to the iris. The donor sclera covered the tube in all cases to decrease the erosion risk. In the Baerveldt implantation procedures, the silicone tube was occluded with an 8-0 absorbable Vicryl suture to minimize the risk of early postoperative hypotony. Sherwood slits were created in the tube with a 10-0 nylon needle to reduce the frequency of early postoperative IOP elevation.

### 2.5. Statistical Analysis

For statistical analysis, a log-rank test was used to determine the association between TLO outcomes and (i) the stage of anterior segment dysgenesis and (ii) corneal staphyloma. The Cochran–Armitage trend test was used to assess the association between glaucoma development and the severity of anterior segment dysgenesis. The Mann–Whitney U test was used to analyze the association between glaucoma and visual outcomes. Statistical significance was set at *p* < 0.05. All statistical analyses were performed using the JMP version 14 statistical package program (Cary, NC, USA).

## 3. Results

### 3.1. Patient Characteristics and Peters Anomaly Diagnosis

Thirty-one eyes of twenty patients were diagnosed with Peters anomaly. The median age at initial presentation was 0.88 years (interquartile range (IQR): 0.053–2.9 years), and the median follow-up time was 8.1 years (IQR: 5.6–9.1 years). The bilateral to unilateral ratio was 11:9, and seven patients were diagnosed with Peters plus syndrome (Table 1), including two with agenesis of the corpus callosum, one with cardiac disease, two with developmental disorders, one with trisomy 13, and one with congenital deafness. Other congenital ophthalmic anomalies include three corneal staphylomas, one persistent fetal vasculature (PFV), and two aniridias.

### 3.2. Association of Glaucoma Development with Severity of Anterior Segment Dysgenesis and Other Congenital Ophthalmic Anomaly

Sixteen eyes of ten patients developed glaucoma secondary to Peters anomaly. Of the sixteen eyes with glaucoma, none, thirteen, and three were classified as Stages 1, 2, and 3 Peters anomaly, respectively. Of the 15 eyes without glaucoma, 2, 12, and 1 were classified as Stages 1, 2, and 3 Peters anomaly, respectively. Stage 1 Peters anomaly had no glaucoma, 52% of Stage 2 had glaucoma, and 75% of Stage 3 had glaucoma. Secondary glaucoma did not occur in the eyes with Stage 1 Peters anomaly, and the incidence of secondary glaucoma tended to increase as the disease progressed (*p* = 0.052; Table 2). All six eyes with other congenital ophthalmic anomalies, including three corneal staphylomas, one PFV, and two aniridias, developed glaucoma. All three eyes with corneal staphyloma were Stage 2.

### 3.3. Association between Glaucoma and Visual Outcomes

Regarding the final VAs, the eyes with glaucoma had the highest percentage of no light perception, and more than half had VA below the finger counting level. In contrast, the non-glaucoma group had the highest percentage of VA better than 20/200, and no cases of blindness were observed (Table 3). The final median VA was 1.8 logMAR (IQR: 1.4–2.0 logMAR) and 1.3 logMAR (IQR: 0.64–1.8 logMAR) in the glaucoma and non-glaucoma groups, respectively. VA was poorer for the eyes with glaucoma than for those without glaucoma (*p* = 0.0292; Table 3).

### 3.4. External Trabeculotomy Outcomes

Of the sixteen eyes with glaucoma, two had good IOP control with medication, eleven underwent glaucoma surgery, and three were ineligible for surgery due to poor general health or ocular conditions (Figure 2). Postoperatively, eight eyes (73%) had well-controlled IOP until the final visit. Among these eyes, three (two underwent TLO and one underwent Ahmed implantation) achieved good IOP control with a single surgery, while five required multiple surgeries. The mean number of surgeries performed was 2.1 ± 1.4 (range 1–5).

TLO was the initial surgery for nine eyes, and five achieved IOP control with TLO performed once or multiple times. Of the five eyes that achieved IOP control by TLO, three were classified as Stage 2 Peters anomaly, and two as Stage 3 Peters anomaly. In contrast, all four eyes that did not achieve IOP control with TLO alone were classified as Stage 2 Peters anomaly (*p* = 0.212). None of the five eyes that achieved IOP control with TLO alone had corneal staphyloma, whereas, among the four eyes where TLO was ineffective, three had corneal staphyloma (*p* = 0.0331). Among the three eyes with poor postoperative outcomes, two developed endophthalmitis after GDD implantation, and one did not achieve IOP control despite multiple surgeries.

## 4. Discussion

Approximately half of patients with Peters anomaly develop glaucoma, and glaucoma secondary to Peters anomaly is an important complication that can affect visual prognosis [6,7,10]. In this study, VA was significantly poorer for the eyes with glaucoma than for those without glaucoma. However, there are few reports on the visual prognosis and treatment outcomes for Peters anomaly with glaucoma. Furthermore, to the best of our knowledge, no studies have specifically investigated the relationship between the severity of anterior segment dysgenesis and the incidence of glaucoma in patients with Peters anomaly. This study investigated the relationship between the incidence of glaucoma and the stage of Peters anomaly and found that the incidence of glaucoma tended to increase as the severity of anterior segment dysgenesis progressed.

Adhesion between the lens and the cornea may occur as the severity of anterior segment dysgenesis increases. Previous studies said that the mechanism of glaucoma onset in Peters anomaly involves trabecular meshwork abnormalities or angle closure due to anterior segment adhesions. This result supported that a consequent anterior shift of the lens can cause angle closure, which may be involved in the development of glaucoma. Glaucoma secondary to Peters anomaly often develops in infancy, and all patients in this study were diagnosed with glaucoma on their initial visits. However, previous reports [17,18] have shown that glaucoma can develop during childhood or later. Therefore, patients with severe anterior segment dysgenesis, particularly those with Stage 3 disease, should be carefully monitored for glaucoma onset.

In the existing literature, there are two reports on the treatment outcomes of glaucoma in Peters anomaly. Dolezal et al. [19] reported that 34 of 58 eyes with Peters anomaly developed glaucoma, and 20 of the 34 eyes underwent glaucoma surgery. Fifteen of the twenty eyes (75%) achieved good IOP postoperatively. Twelve of these fifteen eyes achieved IOP control with GDD implantation. In contrast, additional surgery was required in all eight cases with TLO due to poor IOP control in the early postoperative period. Yang et al. [17] showed that 12 of 34 eyes (32%) with Peters anomaly that underwent glaucoma surgery achieved IOP control. The success rate of the initial surgery was two of seven eyes (28.6%) for TLO and four of four eyes (100%) for GDD. In the present study, of the 16 eyes with glaucoma, glaucoma surgery was performed in 11 eyes, and 8 (73%) achieved good IOP control. The first-line surgery was mainly TLO, and five of nine eyes achieved good IOP control with TLO alone. Our institution has performed many TLOs for childhood glaucoma, not only in Peters anomaly, and experienced doctors performed all the procedures in the present study. However, there were cases in which TLO was difficult to perform. Kupfer and Heath [7,9] revealed that the structures of the anterior chamber angle were grossly normal, aside from evidence of premature aging based on abundant broad-banded collagen fibers in the trabecular lamellae. However, Shields noted that a high insertion of the anterior uvea into the trabecular meshwork is a common finding in some cases of secondary childhood glaucoma [20]. There was a case within this report in which the iris was highly inserted; TLO was attempted, but the trabeculotome came out from behind the iris when the trabeculotome was rotated. TLO may be difficult in cases where the iris covers the trabecular meshwork. After TLO fails to achieve IOP control, GDD implantation or TLE with mitomycin is the treatment of choice. Tanimoto et al. suggested that GDD is the safest procedure to consider after failed conventional angle surgery for refractory childhood glaucoma [21]. According to Malik et al., GDD implantation is associated with 5-year success rates of over 70% in primary childhood glaucoma, and trabeculectomy remains an effective procedure in compliant patients in whom multiple postoperative examinations under anesthesia are possible [22]. Aqueous gels formulated using hydrophilic polymers (hydrogels) and those based on stimuli-responsive polymers (in situ gelling or gel-forming systems) have attracted increasing interest in the treatment of several eye diseases [23]. Improved drug delivery systems may inhibit fibroblast proliferation and yield better surgical outcomes in TLE with mitomycin and GDD implantation. 

In childhood glaucoma, angle surgeries, such as goniotomy and TLO, are considered first-choice surgical procedures, with high success rates of 70–95% reported for primary pediatric glaucoma [24,25]. In contrast, for secondary childhood glaucoma, the success rates of goniotomy and TLO are reported to be lower [21,26]. A previous study reported a lower success rate for angle surgery in patients with primary childhood glaucoma with enlarged ocular dimensions (corneal diameter >14 mm) [27]. In a previous study of the long-term visual prognosis of Peters anomaly and the prognostic factors, eyes with severe disease defined as the presence of other congenital ophthalmic anomalies, corneal staphyloma, microphthalmia, PFV, and aniridia, had a worse prognosis [28]. In the present study, patients with corneal staphyloma had poorer outcomes with TLO. During eye enlargement due to elevated IOP, the structure of the aqueous outflow tract may change. Furthermore, even if the trabecular meshwork and Schlemm canal are adequately opened in TLO, the Schlemm canal may not function properly. In previous reports [17,19], corneal staphyloma was not considered when evaluating surgical outcomes. Therefore, comparing the effect of corneal staphyloma on the outcomes of TLO between this study and previous studies is difficult. However, based on the results of this study, corneal staphyloma was more closely related to the outcomes of TLO than to the stage of anterior segment dysgenesis.

The limitations of this study include its retrospective design, single-center setting, small sample size, and variable follow-up time. Moreover, there was no criterion in selecting the surgical technique for reoperation after the first TLO. The disease severity in this study was classified into Stages 1 through 3, but other papers used types 1 and 2, making it somewhat difficult to compare results because of the difference in severity classification. However, this study also has strengths. A single expert doctor specializing in pediatric glaucoma was involved in diagnosing and treating all cases, ensuring consistency in surgical procedures.

## 5. Conclusions

In conclusion, Peters anomaly cases tend to develop glaucoma as anterior segment dysgenesis progresses. Therefore, careful follow-up of patients with Peters anomaly is necessary, especially those in Stage 3. TLO is effective for glaucoma secondary to Peters anomaly; however, its effectiveness may be limited to cases without corneal staphyloma.

## Figures and Tables

**Figure 1 jcm-12-06238-f001:**
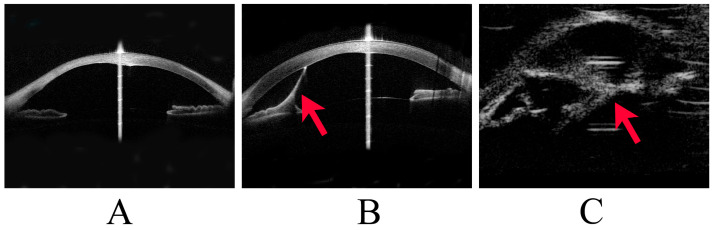
Stages of Peters anomaly. This image shows representative cases of Stage 1 to Stage 3 Peters anomaly. (**A**) Stage 1 Peters anomaly, with anterior segment optical coherence tomography (OCT) showing a posterior corneal defect. (**B**) Stage 2 Peters anomaly, with anterior segment OCT showing a strand (arrowhead) from the iris to the cornea and a posterior corneal defect. (**C**) Stage 3 Peters anomaly, with ultrasound biomicroscopy showing a strand (arrowhead) between the lens and the cornea.

**Figure 2 jcm-12-06238-f002:**
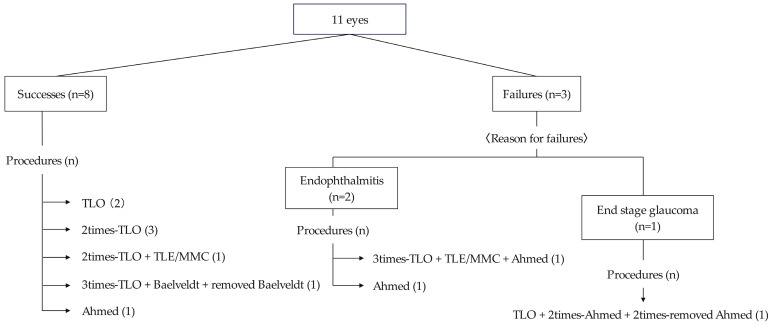
Summary of glaucoma surgeries in this study.

**Table 1 jcm-12-06238-t001:** Patient demographics.

No. of Eyes, Patients	31, 20
Median age at presentation, y (IQR)	0.88 (0.053–2.9)
Median follow-up, y (IQR)	8.1 (5.6–9.1)
Male to female ratio	2:18
Bilateral to unilateral ratio	11:9
Peters plus syndrome, patients	7

IQR: interquartile range.

**Table 2 jcm-12-06238-t002:** Association between glaucoma onset and stage of Peters anomaly.

	Overall Eyes	Eyes with Glaucoma	Eyes without Glaucoma
Stage 1	2	0	2
Stage 2	25	13	12
Stage 3	4	3	1

**Table 3 jcm-12-06238-t003:** Association between glaucoma and final visual acuity.

	Eyes with Glaucoma (%)	Eyes without Glaucoma (%)
20/200≥	2 (15)	4 (40)
20/200<	3 (23)	3 (30)
HM, CF	3 (23)	3 (30)
LP	1 (8)	0
NLP	4 (31)	0

HM: hand motion, CF: counting fingers, LP: light perception, NLP: no light perception.

## Data Availability

Data are contained within the article.

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
