# Peer review of "Risk and Prognostic Factors for Glaucoma Associated with Peters Anomaly"

_jcm, 2023, doi:10.3390/jcm12196238_

Round 1
Reviewer 1 Report
This study aimed to investigate the characteristics of glaucoma asso-ciated with PA and glaucoma surgery outcomes. The group is fairly small, but the topic discussed is of interested.
Here my comments:
- author should insert at least one citation in the introduction on the correlation between PA and glaucoma
- authors should cite the new techniques of treatment of glaucoma, for instance "Fea AM, Novarese C, Caselgrandi P, Boscia G. Glaucoma Treatment and Hydrogel: Current Insights and State of the Art. Gels. 2022 Aug 17;8(8):510. doi: 10.3390/gels8080510. PMID: 36005112; PMCID: PMC9407420."
- the sentence " This study defined successful glaucoma surgery as a postoperative IOP <21 mmHg or Tn (normal tension by palpation) without ocular complications that could affect visual function." shouldn't be ink the statistical analysis section.
- I con't find in the article the limitations discussion
- English grammar should be revised by a native-lenguage
Author Response
The authors thank the reviewers for their kind revision and constructive comments on
our manuscript. We have carefully addressed the comments and made extensive revisions
to our manuscript according to the reviewers’ comments. Our point-by-point responses
to the comments are presented below.
【Response to comments made by Reviewer #1】
- author should insert at least one citation in the introduction on the correlation between PA and glaucoma.
→ We have added details about the correlation between PA and glaucoma with some citations in the introduction section. (Page 1-2, Line 44-56)
- authors should cite the new techniques of treatment of glaucoma, for instance "Fea AM, Novarese C, Caselgrandi P, Boscia G. Glaucoma Treatment and Hydrogel: Current Insights and State of the Art. Gels. 2022 Aug 17;8(8):510. doi: 10.3390/gels8080510. PMID: 36005112; PMCID: PMC9407420."
→ We have cited the provided reference in the discussion section. (Page 8, Line 253-257)
- the sentence " This study defined successful glaucoma surgery as a postoperative IOP <21 mmHg or Tn (normal tension by palpation) without ocular complications that could affect visual function." shouldn't be ink the statistical analysis section.
→ We clarified the definition of successful glaucoma surgery in section 2.3. “Glaucoma diagnosis, Postoperative evaluation“ (Page 3, Line 111-113).
- I can't find in the article the limitations discussion
→ We have added the limitations to the discussion section. (Page 8-9, Line 278-285)
- English grammar should be revised by a native-language
→ We asked a native English speaker to revise our paper.
Reviewer 2 Report
1- Pathogenies of the secondary glaucoma in PA may be included added to the introduction.
2- Lines 95- was the tube inserted through a short tunnel?
3- Any intraoperative difficulty during Trabeculotomy? difficulty in identifying Schlemm’s canal? SC might be absent in some cases.
4- Lines 178-179: do the authors mean" the onset or the incidence of glaucoma" ?
Author Response
The authors thank the reviewers for their kind revision and constructive comments on
our manuscript. We have carefully addressed the comments and made extensive revisions
to our manuscript according to the reviewers’ comments. Our point-by-point responses
to the comments are presented below.
【Response to comments made by Reviewer #2】
1- Pathogenies of the secondary glaucoma in PA may be included added to the introduction.
→ We have added the assumed mechanisms of glaucoma secondary to PA in the introduction section. (Page 1-2, Line 44-53)
2- Lines 95- was the tube inserted through a short tunnel?
→ Yes it was. The GDD tube was inserted approximately 2 mm from the limbus through a short tunnel. (Page 4, Line 127-129)
3- Any intraoperative difficulty during Trabeculotomy? difficulty in identifying Schlemm’s canal? SC might be absent in some cases.
→ In this study, although we could identify Schlemm’s canal in all cases, we encountered a difficult case. There was a case within this report in which the iris was highly inserted; TLO was attempted, but the trabeculotome came out from behind the iris when the trabeculotome was rotated. TLO may be difficult in such cases in which the iris covers the trabecular meshwork. (Page 8, Line 236-246)
4- Lines 178-179: do the authors mean" the onset or the incidence of glaucoma" ?
→ We mean the incidence of glaucoma.
Round 2
Reviewer 1 Report
Authors revisions are proper. No further comments